# Preoperative Halo Traction Versus Direct Posterior Fusion in Severe Adolescent Idiopathic Scoliosis: A Comparative Study

**DOI:** 10.3390/jcm15010142

**Published:** 2025-12-24

**Authors:** Mihai Bogdan Popescu, Harun Marie, Alexandru Ulici, Sebastian Nicolae Ionescu, Mihai Codrut Dragomirescu, Cristiana Popescu, Alexandru Herdea

**Affiliations:** 111th Department of Pediatric Orthopedics, Carol Davila University of Medicine and Pharmacy, Bd. Eroii Sanitari Nr. 8, 050474 Bucharest, Romania; mihai-bogdan.popescu@drd.umfcd.ro (M.B.P.); alexandru.ulici@umfcd.ro (A.U.); alexandru.herdea@umfcd.ro (A.H.); 2Pediatric Orthopedics Department, Grigore Alexandrescu Children’s Emergency Hospital, 011743 Bucharest, Romania; harun_marie@spitalulgrigorealexandrescu.ro (H.M.); mcodrutdragomirescu@spitalulgrigorealexandrescu.ro (M.C.D.); cristiana.popescu0920@stud.umfcd.ro (C.P.); 3Emergency Hospital for Children Maria Sklodowska Curie, 075534 Bucharest, Romania

**Keywords:** halo-gravity traction, adolescent idiopathic scoliosis, spinal deformity correction, posterior spinal fusion, preoperative traction, scoliosis surgery

## Abstract

**Background/Objectives:** This study aimed to evaluate the effect of preoperative halo-gravity traction (HGT) on surgical outcomes in adolescents with severe idiopathic scoliosis (AIS), comparing posterior spinal fusion (PSF) performed with versus without traction in terms of curve correction, complication rates, and overall surgical efficacy. **Methods:** A retrospective cohort study was conducted on 46 adolescents (mean age 14.6 ± 1.9 years) with severe AIS (Cobb > 65°) treated at a single tertiary center between 2011 and 2024. Sixteen patients underwent primary PSF, and 30 received preoperative HGT followed by PSF. Radiographic parameters—including Cobb angle and Risser grade—were analyzed pre- and postoperatively. Statistical tests (*t*-test, Mann–Whitney U, and multivariable linear regression) assessed differences in correction and predictors of outcome, with *p* < 0.05 considered significant. **Results:** Baseline characteristics were comparable between groups (mean preoperative Cobb: 83.6° ± 11.2° vs. 83.1° ± 15.6°, *p* = 0.91). The traction cohort achieved significantly smaller postoperative Cobb angles (30.9° ± 7.8° vs. 42.7° ± 18.9°, *p* = 0.027), greater absolute correction (52.7° ± 7.4° vs. 40.4° ± 10.5°, *p* < 0.001), and higher percentage correction (63.3% ± 6.7% vs. 50.0% ± 14.0%, *p* = 0.002). Regression analysis confirmed HGT as an independent predictor of improved correction (+14.6%, 95% CI +6.9–22.3%, *p* = 0.00047). No neurological or major complications occurred, and most correction was achieved within the first three weeks of traction. **Conclusions:** Preoperative halo-gravity traction significantly enhances deformity correction and surgical safety in severe AIS without added morbidity. Most benefit occurs within 21 days, supporting shorter, standardized traction protocols. HGT remains a valuable adjunct for optimizing outcomes in rigid scoliosis prior to posterior spinal fusion.

## 1. Introduction

Surgical treatment of scoliosis has progressed from the uniplanar distraction era of Harrington rods to contemporary, segmental, three-column strategies that deliver powerful three-dimensional correction with improved safety and durability. Harrington’s distraction rod—widely adopted from the 1950s to 1960s—introduced in-body stabilization and made routine operative fusion possible, but its uniplanar mechanics and poor restoration of sagittal contour produced long-term sequelae (e.g., “flatback”) that motivated further innovation [1]. In the 1970s, Luque’s segmental sublaminar wiring increased points of fixation and internal stability, and in the mid-1980s, Cotrel–Dubousset hook-and-rod systems added rod-rotation and segmental techniques that enabled true three-dimensional derotation and better sagittal control [2]. From the late 1980s onward, pedicle-screw-based constructs supplanted many hook/hybrid; comparative studies and meta-analyses have demonstrated superior coronal correction, lower rates of pseud arthrosis or implant failure and—often—shorter fusion lengths with all-screw constructs [3].

For severe, rigid deformities, techniques have expanded to include powerful osteotomies such as pedicle-subtraction osteotomy and vertebral column resection. These allow multiplane realignment but carry higher complication rates and are therefore frequently combined with staged strategies or preoperative halo-gravity/halo-femoral traction to improve flexibility, pulmonary function and perioperative safety [4]. Contemporary deformity spinal fusion blends segmental pedicle fixation, intraoperative neuromonitoring, image guidance/robotics and tailored osteotomies to maximize three-dimensional correction while minimizing neurologic and cardiopulmonary risk—a clear trajectory of iterative refinement from Harrington’s original concept to today’s multi-modal reconstructive strategies [5].

Posterior spinal fusion (PSF) remains the standard surgical approach for moderate to severe scoliosis, offering biomechanically robust control and consistent deformity correction [6,7]. Modern PSF relies extensively on pedicle screw instrumentation, which delivers superior three-dimensional deformity control compared to hook or hybrid-based constructs [6,8]. In adolescent idiopathic scoliosis (AIS) correction rates of approximately 60–70% are routinely reported using all pedicle screw PSF [9,10]. By avoiding anterior approaches, PSF typically results in shorter operative time, reduced blood loss, and better preservation of pulmonary function, particularly in flexible to moderately rigid curves [6,10,11]. However, in patients with severe rigidity, PSF alone can pose increased neurological and mechanical risks intraoperatively [7,12].

Preoperative halo cranial traction—including halo gravity traction (HGT), halo pelvic or halo femoral traction—can be employed to enhance curve flexibility and reduce surgical complexity in severe rigid scoliosis [6,13]. Meta-analyses and cohort studies confirm that HGT achieves meaningful coronal and sagittal correction (approximately 20–40%) while enhancing pulmonary function, with minimal complications when traction duration is limited to a few weeks and limited additional benefit with extended traction beyond three months [6,7]. In addition, a recent study reported that approximately 85% of Cobb and kyphosis correction occurs within the first three weeks of traction, with minimal further gains thereafter [13]. HGT facilitates safer PSF and may decrease the need for complex osteotomies in rigid deformities [6].

In skeletally immature patients, growth-friendly surgical strategies aim to control deformity while preserving spinal and thoracic growth. Traditional growing rods, implemented as single or dual distraction constructs, allow for serial lengthening at 6–8-month intervals but are associated with high complication rates, including anchor failure, rod fracture, infection and unintended fusion [14]. Magnetically controlled growing rods (MCGR) represent an important advancement by enabling non-invasive outpatient lengthening through external control, thereby reducing the frequency of surgical procedures and repeated anesthetic exposure [15,16]. Meta-analyses and clinical series show that MCGR yields significant Cobb angle reduction and improved spinal length (e.g., T1–S1), although complications such as foundation failure, mechanical distraction loss, and implant fatigue remain non-negligible [15,16]. Besides distraction-based techniques, vertebral hemiepiphysiodesis (e.g., tethering of convex growth plates) seeks gradual guided correction by modulating asymmetric vertebral growth; initial results are promising in well-selected curves, though long-term durability and indications remain under investigation [17].

Although PSF with pedicle screw instrumentation remains the surgical foundation for scoliosis correction, preoperative halo cranial traction presents as a valuable adjunct in selected patients with severe, rigid curves to optimize flexibility, pulmonary reserve, and surgical safety. This study aims to directly compare outcomes of PSF performed with versus without preoperative HGT, focusing on curve correction, complication rates and overall surgical efficacy.

## 2. Materials and Methods

### 2.1. Study Design

This investigation was designed as a retrospective cohort analysis aimed at evaluating the role of preoperative halo-gravity traction in modifying curve severity and improving surgical outcomes in severe scoliosis. Results from this group were contrasted with those from patients who underwent primary posterior spinal fusion without traction. The research protocol was approved by the institutional ethics committee on 14 March 2025 (approval no. 9). For all participants, written informed consent was obtained from parents or legal guardians. Data were retrieved from the clinical records of children treated in a single tertiary orthopedic center between 2011 and 2024.

### 2.2. Participants

The study population consisted of adolescents diagnosed with idiopathic scoliosis (AIS), organized into two matched cohorts. AIS was defined as a coronal spinal deformity with a Cobb angle exceeding 65°. The first cohort included patients treated between 2011 and 2018 with posterior spinal fusion only, whereas the second cohort included those treated between 2018 and 2024 who received halo-gravity traction prior to spinal fusion. Patients with congenital malformations, neuromuscular etiologies, post-traumatic scoliosis or incomplete data sets were excluded.

### 2.3. Study Procedure

In the first cohort, posterior instrumented spinal fusion was carried out using an approach that combined vertebral fusion with segmental fixation employing pedicle screws, laminar hooks and dual rods. The choice of fusion levels depended on curve morphology and flexibility, with the objective of maximizing correction while sparing mobile segments when feasible. Ponte osteotomies were performed selectively to enhance spinal flexibility and restore sagittal alignment. All interventions were performed under general anesthesia by surgeons specialized in scoliosis management. The mean surgical duration was approximately 6 h, and patients generally remained hospitalized for 1 week postoperatively, until independent ambulation was achieved.

In the second cohort, halo traction was implemented before surgery. A halo ring was applied under general anesthesia and secured with four titanium pins. Gradual axial traction was initiated through a pulley-and-weight system, beginning without load and increased daily by 1 kg until reaching 40–50% of the individual’s body weight. The traction apparatus allowed mobilization between bed, wheelchair, and walker (Figure 1).

Patients were assessed daily for potential complications such as pin site infections or neurological symptoms, with adjustments to the load according to clinical tolerance and radiographic feedback. After the traction phase, all patients underwent posterior spinal fusion with instrumentation.

The inclusion criteria required a minimum follow-up of 12 months, complete clinical and radiological data, and informed consent to participate in the study. Standing anteroposterior and lateral full-spine radiographs were obtained preoperatively and postoperatively at 2 weeks, 1 month, 3 months, 6 months and annually thereafter. This is exemplified in Figure 2 and Figure 3.

The primary outcome measured was postoperative improvement of the Cobb angle. For patients treated with halo-gravity traction before surgery, an additional parameter was evaluated: the percentage reduction in the Cobb angle that occurred during the traction period itself. Skeletal maturity was determined from pelvic radiographs using the Risser grading system. To ensure consistency, Cobb angle measurements were independently performed by two orthopedic surgeons and in cases of discrepancy greater than 5°, the average of the two values was used.

### 2.4. Statistical Analysis

For each patient, demographic and clinical data such as age, sex, body weight, traction load, and duration of traction were recorded. Statistical analysis was conducted using SPSS version 26.0 (IBM Corp., Armonk, NY, USA). Continuous variables (age, Cobb angle, Risser score) were summarized with means, standard deviations, and ranges. Paired *t*-tests were applied to compare baseline and postoperative Cobb angles, thereby quantifying surgical correction.

The specific contribution of halo traction was analyzed by examining postoperative correction in relation to pre- and post-traction changes. Correlation analyses (Pearson’s r) explored the association of postoperative improvement with age and Risser grade. Differences in Cobb angle correction across Lenke curve classifications were examined with the Kruskal–Wallis test. Linear regression models were further used to identify predictors of postoperative outcome, including age, baseline Cobb angle and skeletal maturity. A significance threshold of *p* < 0.05 was adopted for all analyses.

## 3. Results

A total of 46 patients that meet the inclusion criteria were selected: 30 with traction and 16 without traction. Demographic data, including gender distribution and Cobb angle, are presented in Table 1.

Both groups were comparable in baseline demographic and deformity parameters.

The mean age was 14.5 ± 1.9 years in the traction group and 14.8 ± 1.9 years in the non-traction group, with no statistically significant difference (*p* = 0.91). Sex distribution differed, with the traction cohort consisting predominantly of females (23 females, 7 males), while the non-traction group contained mostly females as well (15 females, 1 male). The initial Cobb angle was almost identical between groups—83.6° ± 11.2° for the traction group and 83.1° ± 15.6° for the non-traction group—with *p* = 0.91, confirming that both groups started with comparable curve magnitudes prior to traction or surgery.

Preoperatively, the mean starting Cobb angle was 83.6° ± 11.2° in the traction group and 83.1° ± 15.6° in the no-traction group (Welch’s t = 0.11, *p* = 0.91; Mann–Whitney U = 284.0, *p* = 0.315), indicating virtually identical baseline deformities. Postoperatively, the traction group achieved significantly smaller curves (30.9° ± 7.8° vs. 42.7° ± 18.9°; *p* = 0.027), with an average 12° greater residual correction. The absolute reduction was also significantly higher in the traction cohort (52.7° ± 7.4° vs. 40.4° ± 10.5°; *p* < 0.001), corresponding to a mean gain of roughly 13° more correction. Similarly, the percentage correction was 63.3% ± 6.7% with traction compared to 50.0% ± 14.0% without traction (*p* = 0.002), representing a 13% higher proportional correction. Within the traction cohort, extending treatment beyond three weeks offered no measurable benefit: patients treated ≤ 21 days achieved 63.7%, while those > 21 days achieved 63.2% (*p* = 0.959).

The percentage of correction was also superior in the traction cohort (63.3% ± 6.7% vs. 50.0% ± 14.0%; Welch’s t = 3.60, *p* = 0.002; Mann–Whitney U = 389.0, *p* = 0.0006). This represents an additional 13% proportional correction in the traction group, as seen in Table 2.

Following treatment, the traction group achieved significantly smaller postoperative Cobb angles compared to the non-traction group (30.9° ± 7.8° vs. 42.7° ± 18.9°, *p* = 0.027). The absolute Cobb angle reduction was also greater in the traction cohort (52.7° ± 7.4° vs. 40.4° ± 10.5°, *p* < 0.001). Similarly, the percentage of correction was higher with traction (63.3% ± 6.7%) than without traction (50.0% ± 14.0%, *p* = 0.002).

Within the traction cohort 6 patients treated for ≤21 days achieved a mean percentage correction of 63.7%, while 24 patients treated for >21 days achieved 63.2%. This difference was not statistically significant (Mann–Whitney U = 73.5, *p* = 0.959). Most correction occurred within the first three weeks and prolonging traction beyond 21 days offered no measurable advantage. During the traction period, cervical pain was reported by the majority of patients, 80%, typically mild to moderate in intensity and responsive to oral or topical anti-inflammatory therapy. Six patients developed superficial pin-site infections, all of which resolved without systemic intervention. Other transient adverse effects included headache, pin discomfort, mild sensory changes in the upper extremities—promptly relieved by reducing traction weight by 1 kg—dizziness and backache.

The small sample size of patients with Lenke types 1, 2, 3, and 5 curves in one or both study groups precluded a reliable intergroup comparison. Among Lenke 4 patients, traction produced higher mean percentage correction than no traction, but the difference was not statistically significant (Welch’s t = 1.60, *p* = 0.14; Mann–Whitney U = 5.0, *p* = 0.20). Traction tended to improve correction across Lenke types, but sample sizes were too small for firm conclusions.

As shown in Figure 4, postoperative Cobb angles were significantly lower in the traction group compared to the no-traction group (30.9° ± 7.8° vs. 42.7° ± 18.9°; *p* = 0.027). Most patients in the traction cohort clustered tightly between 25° and 35°, indicating a more consistent correction profile, while the non-traction group displayed greater variability with several residual curves exceeding 50°. This visual distribution highlights the stabilizing and homogenizing effect of preoperative halo traction, which both reduced mean postoperative deformity and limited outliers with poor correction.

As illustrated in Figure 5, the mean proportional correction was significantly higher in the traction cohort (63.3% ± 6.7%) than in the non-traction cohort (50.0% ± 14.0%; *p* = 0.002). The reduced dispersion within the traction group suggests a more uniform and predictable corrective effect, in contrast to the broader variability observed in the non-traction group.

In multivariable linear regression adjusting for age, Lenke classification and Risser stage, traction remained an independent predictor of greater percentage correction (ẞ = +14.6%, 95% Cl +6.9% to +22.3%, *p* = 0.00047). Neither age nor Risser stage was significantly associated with percentage correction, and no Lenke subtype effect reached significance.

## 4. Discussion

Despite the evolution of powerful segmental instrumentation and advanced corrective techniques, halo-gravity traction (HGT) remains a vital adjunct in the modern surgical management of severe and rigid scoliosis. Contemporary deformity correction relies not only on implant strength but also on the gradual mobilization of stiff curves to minimize neurologic and cardiopulmonary risk. HGT offers a unique, physiologic means of progressive correction that enhances spinal flexibility, optimizes pulmonary function, and allows safer, more controlled intraoperative maneuvers. In an era where complex osteotomies and three-column resections carry substantial morbidity, traction provides a less invasive preparatory step that can reduce surgical demands while improving overall alignment. Its continued relevance reflects a shift towards safer, staged correction strategies that prioritize patient safety, gradual adaptation, and multidisciplinary care [18].

This study demonstrates that preoperative halo-gravity traction (HGT) provides a measurable advantage in the surgical management of severe scoliosis. Among 46 patients with comparable baseline curve magnitudes (83.6° vs. 83.1°), those treated with HGT achieved smaller postoperative Cobb angles (30.9° vs. 42.7°), larger absolute reductions (52.7° vs. 40.4°), and superior percentage correction (63.3% vs. 50.0%). Importantly, multivariable regression confirmed that traction independently contributed to an additional 11% gain in correction, irrespective of age, skeletal maturity or curve type. These results support HGT as an effective adjunct to enhance correction in rigid scoliosis.

Our results align with prior literature demonstrating radiographic and functional benefits of HGT. Multiple series have shown preoperative curve reductions of 20–40%, which translate into improved final surgical correction [7,19,20]. The mean correction percentage of 63% observed in our HGT cohort is consistent with the upper range of values reported in contemporary studies. Importantly, the enhanced correction persisted postoperatively, reinforcing that HGT provides durable advantages rather than only transient preoperative improvements.

We observed that the majority of deformity correction achieved with halo-gravity traction occurred within the first three weeks of treatment, with no significant additional improvement with longer traction duration. This finding suggests a plateau effect, whereby extending traction beyond this initial period yields diminishing radiographic benefit while potentially increasing hospital stay, cost and patient burden. This plateau effect has been described previously, with studies suggesting that prolonged traction beyond 4–6 weeks yields diminishing returns [7,19,20,21]. Shorter traction protocols may therefore maximize benefit while reducing hospital stay, complications and cost.

Subgroup analysis by Lenke classification suggested a consistent, though not statistically significant, advantage of HGT across curve types, particularly in Lenke 4 deformities. Although underpowered, this trend supports earlier observations that traction is broadly beneficial in rigid coronal and sagittal patterns [22]. Larger multicenter cohorts are needed to clarify whether certain curve types derive disproportionate benefit.

The effect of traction was independent of age and skeletal maturity in our series. This finding extends prior work emphasizing curve rigidity and pulmonary compromise as major determinants of traction benefit [7]. It suggests that HGT may add value across a wider range of patients with severe scoliosis, not limited to skeletally immature individuals.

Safety is another important consideration. We observed minimal complications, consistent with systematic reviews showing that HGT is generally safe, with low rates of neurologic events and mostly minor pin-site problems [20,23].

Although pulmonary function was not directly assessed in this study, multiple prior investigations have demonstrated that halo-gravity traction significantly improves ventilatory mechanics and forced vital capacity, thereby optimizing perioperative cardiopulmonary reserve. In addition to coronal plane correction, halo-gravity traction has also been shown to favorably influence sagittal alignment, particularly thoracic kyphosis and global spinal balance [7,19,24,25]. Finally, beyond radiographic and physiologic improvements, patients treated for adolescent idiopathic scoliosis have been shown to achieve significant postoperative gains in health-related quality of life, as measured by validated instruments such as the SRS-30 questionnaire [26]. This supports the concept that preoperative preparatory strategies, including halo-gravity traction, may contribute not only to improved structural correction but also to meaningful patient-centered outcome domains

Due to the retrospective nature of the study several important perioperative and clinical variables—including formal preoperative flexibility radiographs, intraoperative blood loss, detailed osteotomy utilization, operative time, neuromonitoring changes, postoperative pain scores, pulmonary function testing, patient-reported outcome measures and long-term revision rates and complications—were not uniformly available across the entire study period, particularly in the historical non-traction cohort. Nevertheless, the primary study endpoint of radiographic correction was consistently available and objectively measured in all patients.

Another limitation was the modest sample size, particularly for subgroup analyses, which limited statistical power. The retrospective design introduces the possibility of selection bias and the relatively short follow-up precludes assessment of long-term correction durability. Finally, although we adjusted for several confounders, unmeasured variables such as pulmonary function or nutritional status may have influenced outcomes. Nevertheless, the consistent and statistically robust advantage of HGT supports its role as an effective adjunct in severe scoliosis management.

Clinical implications: For patients with rigid curves exceeding 80°, HGT should be considered to enhance correction and potentially reduce the need for extensive osteotomies. Most of the benefit is achieved within the first three weeks, suggesting that extended protocols may not be necessary. Future prospective, multicenter studies with standardized traction protocols and incorporation of pulmonary and patient-reported outcomes will be essential to refine best practices.

## 5. Conclusions

Halo-gravity traction is a safe and effective adjunct for managing severe, rigid scoliosis. It enhances spinal flexibility, improves postoperative alignment, and reduces intraoperative risks. Standardized, evidence-based protocols—ideally limited to about three weeks—can guide reference centers in optimizing results while minimizing hospital stay and complications. Overall, HGT improves surgical outcomes, decreases the need for complex corrective maneuvers and reduces the overall surgical burden for both the patient and the surgeon, making it a valuable component of modern scoliosis management.

## Figures and Tables

**Figure 1 jcm-15-00142-f001:**
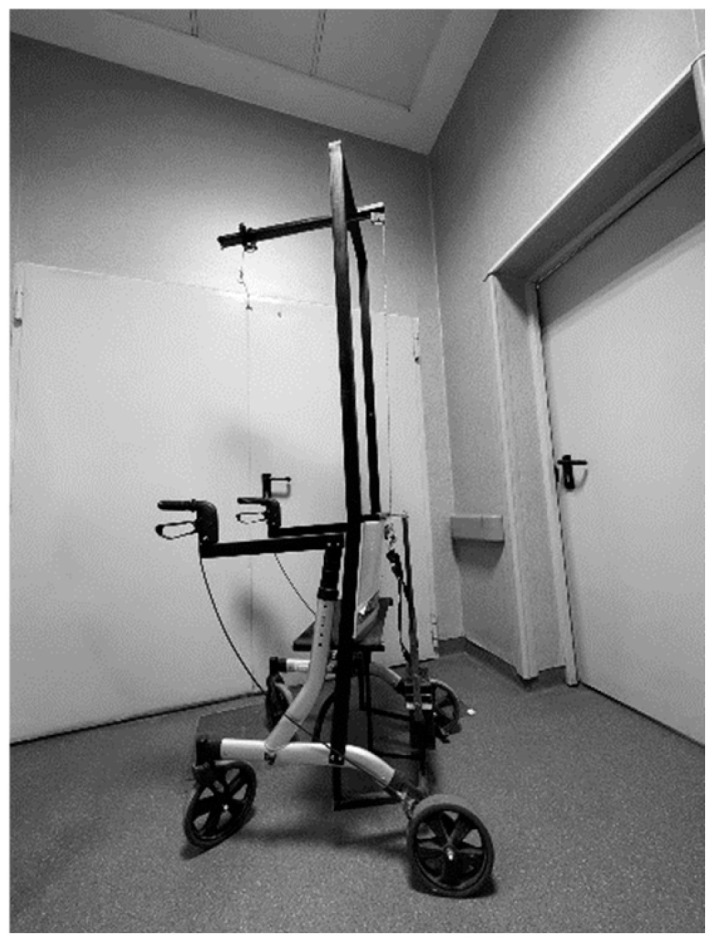
Walker combined with a traction system.

**Figure 2 jcm-15-00142-f002:**
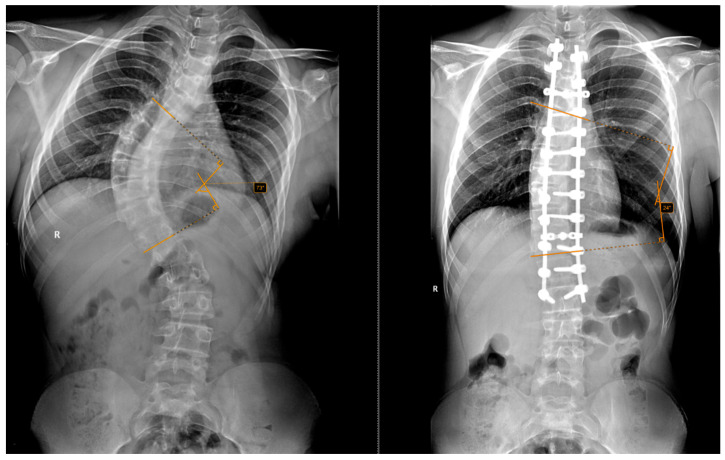
Preoperative and postoperative radiographs of a 14-year-old patient who did not undergo halo-traction. The (**left**) image shows a preoperative Cobb angle of 73° for the main curve (T6–T12), while the (**right**) image shows the postoperative correction to 24°.

**Figure 3 jcm-15-00142-f003:**
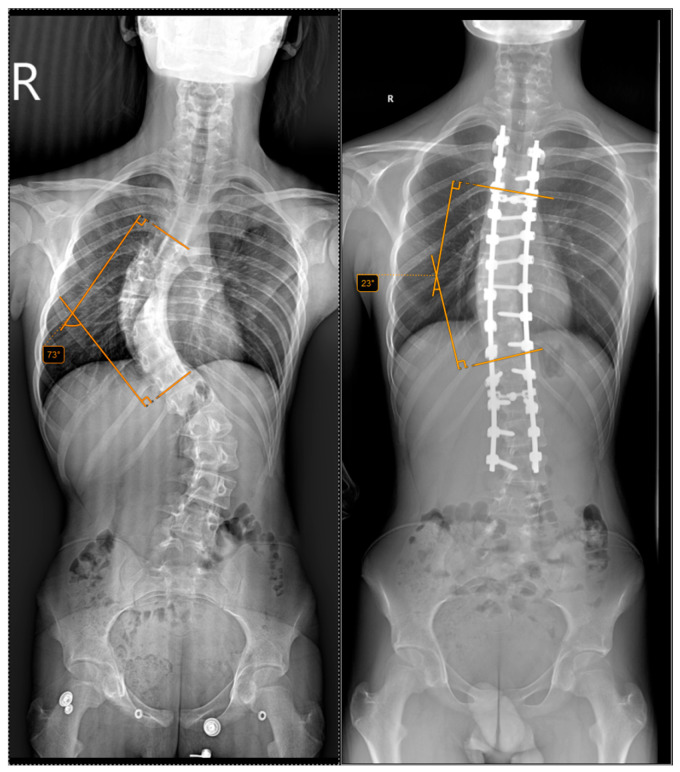
Preoperative and postoperative radiographs of a 13-year-old patient who underwent halo-traction. The (**left**) image shows a preoperative Cobb angle of 73° for the main curve (T5–T11), while the (**right**) image shows the postoperative correction to 23°.

**Figure 4 jcm-15-00142-f004:**
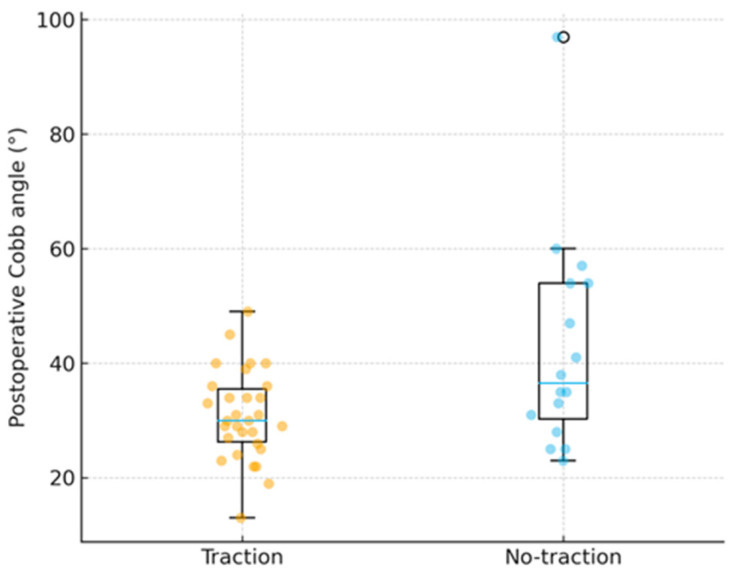
Distribution of postoperative Cobb angles in patients with and without preoperative halo traction. Each dot represents one patient; boxes show interquartile range with median lines and whiskers representing range.

**Figure 5 jcm-15-00142-f005:**
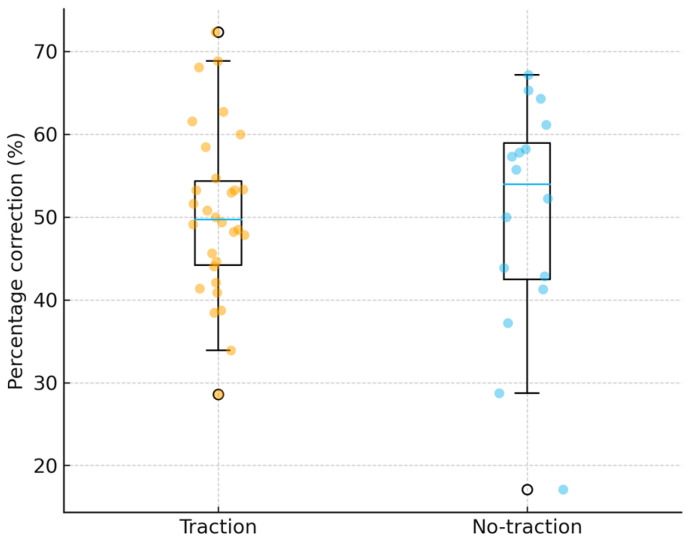
Percentage reduction in major curve magnitude between the starting and postoperative Cobb angles in the traction and no-traction cohorts. Boxes represent interquartile ranges; dots indicate individual patients.

**Table 1 jcm-15-00142-t001:** Baseline characteristics of the traction and no-traction groups, including age, sex distribution, and initial Cobb angle with 95% confidence intervals.

	Traction Group (N = 30)	No-Traction Group (N = 16)
Age (years)	14.5 ± 1.9 (10–17)	14.8 ± 1.9 (11–17)
Gender (F/M)	23/7	15/1
Initial cobb angle (°) [95% CI]	83.6 ± 11.2 (80.0; 87.2)	83.1 ± 15.6 (75.6; 90.5)

**Table 2 jcm-15-00142-t002:** Postoperative outcomes and correction analysis comparing the traction and no-traction groups.

	Traction Group (N = 30)	No-Traction Group (N = 16)
Postoperative cobb angle (°) [95% CI]	30.9 ± 7.8 (28.1; 33.6)	42.7 ± 18.9 (33.7; 51.6)
Absolute reduction (°) [95% CI]	52.7 ± 7.4 (49.9; 55.6)	40.4 ± 10.5 (35.0; 45.8)
Percentage correction (%) [95% CI]	63.3 ± 6.7 (60.7; 65.9)	50.0 ± 14.0 (43.1; 56.9)

## Data Availability

The data presented in this study are available on request from the corresponding author due to privacy and ethical reasons.

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
