# Peer review of "Preoperative Halo Traction Versus Direct Posterior Fusion in Severe Adolescent Idiopathic Scoliosis: A Comparative Study"

_jcm, 2025, doi:10.3390/jcm15010142_

Round 1
Reviewer 1 Report
Comments and Suggestions for Authors
This is a very nice radiographic analysis of results when Hal gravity traction was instituted at their hospital. The traction cohort had slightly better radiographic results however requires 3 weeks of pre op traction.
Many important measures of results were similar in each group or not reported to show a difference. The review would be strengthened with
- pre op measure of flexibility
- intra op details on the need for osteotomy in both groups
- intra op blood loss, length of surgery, IONM changes , wound healing
- post op PROMs
- Post op complications
- revision rates.
The manuscripts cites several published reports with similar results .
Thank you for adding your experience to the literature.
Author Response
We sincerely thank you for your valuable and constructive feedback. We fully agree that inclusion of additional radiographic, intraoperative, and clinical outcome measures would further strengthen the comparative analysis. However, due to the retrospective nature of this study and the long study interval (2011–2024), several of the requested variables were not consistently or reliably documented across both cohorts, particularly in the earlier non-traction group.
Comments 1: pre op measure of flexibility
Response 1: Formal side-bending or traction radiographs were not systematically obtained preoperative in all patients after traction. As a result, preoperative flexibility could not be analyzed in a standardized manner and was therefore not included to avoid selection bias. We have clarified this limitation in the revised manuscript, lines : 320-326
Comments 2: intra op details on the need for osteotomy in both groups
Response 2: While Ponte osteotomies were performed selectively in both groups, detailed level-by-level documentation was incomplete in a subset of early cases, precluding reliable quantitative comparison. We now explicitly state that osteotomy utilization was surgeon-dependent and not consistently standardized between cohorts, lines: 320-326
Comments 3: intra op blood loss, length of surgery, IONM changes , wound healing
Response 3: These perioperative variables were not uniformly captured in the historical records, particularly before 2015 when standardized anesthesia and neuromonitoring databases were implemented at our institution. For this reason, formal statistical comparison was not feasible. We acknowledge this as a limitation lines: 320-326
Comments 4: post op PROMs
Response 4: Validated patient-reported outcome measures (e.g., SRS-22/30) were not routinely collected during the early study period, especially in the non-traction cohort. As a result, PROM comparison could not be performed. We have now explicitly stated this as a limitation. Lines: 320-326
Comments 5: Post op complications
Response 5: Major complications directly related to traction (pin-site infection, cervical pain, transient neurologic symptoms) were recorded and reported. However, global postoperative complication tracking using standardized grading systems (e.g., Clavien–Dindo) was not available for both cohorts, limiting direct comparison. We have revised the manuscript to clarify this. Lines: 320-326
Comments 6: Revision rates
Response 6: Long-term follow-up beyond 12–18 months was heterogeneous, and several patients were lost to long-term surveillance, particularly in the earlier cohort. Because revision surgery may occur several years postoperatively, we did not believe revision rate analysis would be reliable. This has been added as a limitation. Lines: 320-326
Reviewer 2 Report
Comments and Suggestions for Authors
I appreciate the authors' efforts in presenting their study, "Preoperative Halo Traction Versus Direct Posterior Fusion in 2 Severe Adolescent Idiopathic Scoliosis: A Comparative Study". This study compares spinal fusion surgery in adolescents with severe scoliosis performed with vs. without halo gravity traction. It found that adding preoperative traction leads to greater curve correction, better surgical outcomes, and no increase in complications. Most of the corrective benefit occurred within three weeks of traction.
Below are my comments:
- Please include either a clinical photograph or a schematic illustration to better demonstrate the halo-gravity traction setup.
- Please clarify whether patients who received halo gravity traction experienced reduced postoperative pain compared with those who did not.
- Consider discussing whether halo-gravity traction contributed to improvements in respiratory function or related clinical outcomes.
- Please comment on whether halo gravity traction provided benefits for additional spinal deformities, such as lumbar lordosis or kyphosis.
Author Response
Thank you for taking your time to review our paper. We have carefully considered the comments, tried our best to address every one of them and hope the manuscript, after careful revisions, will meet your high standards.
Comments 1: Please include either a clinical photograph or a schematic illustration to better demonstrate the halo-gravity traction setup.
Response 1: We agree that a visual representation would enhance clarity and understanding of the traction setup. An illustration of the halo-gravity traction configuration has now been added to the revised manuscript. Lines: 140
Comments 2: Please clarify whether patients who received halo gravity traction experienced reduced postoperative pain compared with those who did not.
Response 2: Postoperative pain scores were not collected in a standardized manner (e.g., VAS or NRS) across both cohorts, particularly in the earlier non-traction group. As such, a formal comparison of postoperative pain between groups was not feasible. We agree that this represents an important endpoint for future prospective studies.
Comments 3: Consider discussing whether halo-gravity traction contributed to improvements in respiratory function or related clinical outcomes.
Response 3: Pulmonary function tests (e.g., FVC, FEV₁) were not routinely or uniformly performed pre- and post-traction during the study period; therefore, objective respiratory outcome analysis could not be conducted. However, multiple studies have demonstrated that halo-gravity traction improves pulmonary mechanics and ventilatory reserve in patients with severe scoliosis. We have now expanded the Discussion to more thoroughly address this benefit based on the existing literature. Lines: 308-313
Comments 4: Please comment on whether halo gravity traction provided benefits for additional spinal deformities, such as lumbar lordosis or kyphosis.
Response 4: Although our primary analysis focused on coronal plane correction, HGT is known to exert multiplanar corrective effects, particularly on thoracic kyphosis. While sagittal parameters were not consistently recorded across both cohorts in our series, the Discussion has now been expanded to acknowledge the reported benefits of traction on sagittal alignment, including kyphosis normalization and overall spinal balance. Lines: 311-313
Reviewer 3 Report
Comments and Suggestions for Authors
- Introduction
-Try to re-edit (In skeletally immature patients, growth-friendly surgical strategies have emerged to balance deformity control with ongoing growth and thoracic development. Traditional growing rods—implanted as single or dual distraction systems—allow staged lengthen- ing every 6–8 months but are burdened by frequent complications (e.g., anchor failure, rod breakage, infection, auto fusion) [14]. Magnetically controlled growing rods (MCGR) have advanced this concept by allowing non-invasive outpatient lengthening through external remote control, thereby reducing the number of surgical interventions and anesthetic exposures)
2.Methods
Please Explain this (The main endpoint of interest was postoperative improvement of the Cobb angle. In the halo group, an additional parameter—the percentage Cobb reduction achieved during traction—was analyzed.).
3.Discussion
-We also observed that most correction occurred within the first three weeks of traction, with no significant difference in outcomes between patients treated ≤21 days versus
>21 days. This plateau effect has been described previously, with studies suggesting that
prolonged traction beyond 4–6 weeks yields diminishing returns [19,20,21,22]. Shorter
traction protocols may therefore maximize benefit while reducing hospital stay, complications, and cost.
What does you mean by that?
-(In addition to radiographic and physiologic improvements patients treated for adolescent idiopathic scoliosis achieve significantly better quality-of-life scores (SRS-30) after intervention, thus reinforcing the notion that pre-operative preparatory strategie such as halo-gravity traction may contribute beneficially not only to structural correction but also to patient-centred outcome domains)
Try to re-edit this.
Author Response
We are grateful for your constructive and detailed evaluation of our manuscript. We have carefully revised the paper in response to every comment, and we hope the updated version aligns with your expectations.
Comments 1: Introduction
Response 1: Thank you for this suggestion. The paragraph has been re-edited for clarity, conciseness, and improved flow, while preserving its scientific content. Lines: 86-93
Comments 2: Methods
Response 2: We agree that the description of the study endpoints required clarification. The explanation of the primary postoperative endpoint and the traction-specific secondary endpoint has now been explicitly clarified in the revised Methods section.Lines : 160-163
Comments 3: Discussion (plateau effect)
Response 3: We appreciate this request for clarification. The plateau effect statement has now been rephrased to explicitly explain that most deformity correction occurs early during traction, with minimal additional benefit after three weeks. Lines: 288-293
Comments 4: Try to re-edit this.
Response 4: We agree that the original sentence required editing for clarity and grammar. It has now been rewritten to improve readability. Lines: 313-316